# Identification of Immunogenic Antigens of *Naegleria fowleri* Adjuvanted by Cholera Toxin

**DOI:** 10.3390/pathogens9060460

**Published:** 2020-06-10

**Authors:** Saúl Rojas-Hernández, Mara Gutiérrez-Sánchez, Diego Alexander Rojas-Ortega, Patricia Bonilla-Lemus, Arturo Contis-Montes de Oca, Jorge Herrera-Díaz, Israel López-Reyes, María Maricela Carrasco-Yépez

**Affiliations:** 1Laboratorio de Inmunobiología Molecular y Celular, Sección de Estudios de Posgrado e Investigación, Salvador Díaz Mirón esq. Plan de San Luis S/N, Miguel Hidalgo, Casco de Santo Tomas, Ciudad de Mexico 11340, Mexico; srojash@ipn.mx (S.R.-H.); mgutierrezsa@ipn.mx (M.G.-S.); alexanderrojas1589@gmail.com (D.A.R.-O.); 2Laboratorio de Microbiología, Grupo CyMA, Unidad de Investigación Interdisciplinaria en Ciencias de la Salud y la Educación, Universidad Nacional Autónoma de México, UNAM FES Iztacala, Los Reyes Iztacala, Tlalnepantla C.P. 54090, Mexico; blemus@unam.mx (P.B.-L.); neocontis@hotmail.com (A.C.-M.d.O.); 3Unidad de Servicios de Apoyo a la Investigación y la Industria, Facultad de Química, Universidad Nacional Autónoma de México, Ciudad de Mexico C.P. 04510, Mexico; jherredi@unam.mx; 4Universidad Autónoma de la Ciudad de México (UACM), Plantel Cuautepec, Av. La Corona 320, Col. Loma la Palma, Alcaldía Gustavo A. Madero, Ciudad de Mexico C.P. 07160, Mexico; israel.lopez.reyes@uacm.edu.mx

**Keywords:** *Naegleria fowleri*, immunogenic antigens, cholera toxin, vaccine candidates

## Abstract

The intranasal administration of *Naegleria fowleri* lysates plus cholera toxin (CT) increases protection against *N. fowleri* meningoencephalitis in mice, suggesting that humoral immune response mediated by antibodies is crucial to induce protection against the infection. In the present study, we applied a protein analysis to detect and identify immunogenic antigens from *N. fowleri*, which might be responsible for such protection. A Western blot assay of *N. fowleri* polypeptides was performed using the serum and nasal washes from mice immunized with *N. fowleri* lysates, either alone or with CT after one, two, three, or four weekly immunizations and challenged with trophozoites of *N. fowleri.* Immunized mice with *N. fowleri* plus CT, after four doses, had the highest survival rate (100%). Nasal or sera IgA and IgG antibody response was progressively stronger as the number of immunizations was increased, and that response was mainly directed to 250, 100, 70, 50, 37, and 19 kDa polypeptide bands, especially in the third and fourth immunization. Peptides present in these immunogenic bands were matched by nano-LC–ESI-MSMS with different proteins, which could serve as candidates for a vaccine against *N. fowleri* infection.

## 1. Introduction

Primary amebic meningoencephalitis (PAM) is an acute and fatal disease of the central nervous system, whose etiologic agent is *Naegleria fowleri.* As of 2012, 310 cases had been reported globally, with a fatality rate of more than 95% [1]. The amoeba generates a disease that progresses very quickly, since it enters the host through the nasal cavity and invades the brain, generally causing death in 3 to 7 days [2,3]. Those most affected by PAM are healthy children under the age of 13 who have had recent exposure to warm freshwater [4]. 

Experimentally, resistance to infection against *N. fowleri* has been induced in our laboratory hosts, which involves immunizing mice by the intranasal (i.n) route with *N. fowleri* lysates in combination with cholera toxin (CT) or Cry1Ac protoxin as adjuvants [5,6]. The success of this protection is thought to arise from the intranasal route of administration and the use of CT as an adjuvant, which favors the induction of local specific antibody response against *N. fowleri***,** playing an important role in infection control. Mice under this immunization scheme have shown an increase in Th2 immune response, which involved molecules such as IgA, IgG, and IgG1 antibodies, interleukin-4 (IL-4), as well as an increase in the following gene expressions: IgA alpha chain, polymeric Ig receptor (pIgR), and cytokines such as IL-10, IL-6, and IFN-g [6,7]. In another work, using the same immunization scheme, we analyzed T- and B-lymphocyte populations, where it was shown that there was a major number of CD4 rather than CD8 T-lymphocytes, and the IgA antibody-forming cell (IgA-AFC) response was increased mainly in the nasal passage (NP) and nasopharyngeal-associated lymphoreticular tissue (NALT) compartments. Furthermore, macrophages and dendritic cells from NALT, NP, and cervical lymph nodes (CLN) in immunized mice expressed high CD80 and CD86 levels [8]. It was suggested that these factors are involved in the strong presence of specific IgA e IgG antibodies found in the lumen from immunized and protected mice [6]. 

One way to control and eliminate *N. fowleri* infection in endemic areas would be to develop an effective and safe vaccine. In previous studies, a recombinant Nfa1 protein (rNfa1) with a molecular weight of 13.1 kDa, intranasally administered with cholera toxin B subunit (CTB) or the enterotoxigenic *Escherichia coli* heat-labile toxin B subunit (LTB) adjuvants as vaccine strategies for *N. fowleri* infection, has gained attention because splenocytes from the immunized mice secreted Th1 type cytokines (IFN-γ), Th2 type cytokines (IL-4), and regulatory cytokines (IL-2 and IL-10). Those results suggested that the immunization with rNfa1 protein, using CTB and LTB, elicited a Th1/Th2/Treg mixed-type immune response in *N. fowleri*-infected mice, demonstrating that Nfa1 protein is a potential candidate as a vaccine antigen against *N. fowleri* infection [9].

The characterization of proteins responsible for pathogenicity and immunogenicity of *N. fowleri* is still incomplete [10]. In this regard, in a recently published work, we detected by 2-DE Western blot different protein spots between *Naegleria fowleri* (pathogenic amoeba) and *Naegleria lovaniensis* (nonpathogenic amoeba) that were recognized by *N.fowleri*-specific IgG antibodies from protected, immunized mice against the amoeba. The analysis of these protein spots revealed changes in size and intensity between both species. Moreover, mass spectrometry also identified differences in proteins, such as actin fragment, myosin II, heat shock protein, membrane protein Mp2CL5 regarding sequence coverage, and the number of post-translational modifications, that had not been previously reported. [11]. However, the identification of immunogenic antigens recognized by both local and systemic specific antibodies, including a scheme with a different number of immunizations and the use or not of CT as an adjuvant, has not yet been analyzed. Thus, the priority aims of the present work are focused on these last factors.

It has been shown that specific antibodies directed at cell antigens of several microorganisms, including *Neisseria meningitidis* [12], *Streptococcus pneumoniae* [13], and *Haemophilus influenzae* [14], can generate protection against these pathogens that cause experimental disease. These results suggest that it is possible to use these immunogenic antigens, which are strongly recognized by specific IgA e IgG antibodies, as vaccine candidates to control natural infections caused by these microorganisms.

Identification of specific molecules composed of antigens of *N. fowleri* that could be detected in our immunization model and selected by the antibodies responsible for inducing protective humoral response greatly facilitate the selection of promising vaccine candidates for further evaluation. These immunogenic molecules could offer some advantages over immunization with the whole microorganism as they are easier to produce, their effects on the immune response can be delimited more clearly, and they can be free of bacterial or parasite contaminants that may potentially induce negative side effects such as the induction of autoimmunity or toxic effects [15].

Therefore, these findings led our group to attempt to identify vaccinating antigens among the major immunogenic polypeptides recognized of *N. fowler**i* by specific IgA, IgG and IgM antibodies from mice immunized with *N. fowleri* lysates plus CT or *N. fowleri* lysates alone, with a different number of immunizations (1, 2, 3 or 4), and examining whether the survival rate could be related to the recognition of these antigens by the specific antibodies.

## 2. Results

### 2.1. Survival and Protection

Table 1 shows the survival of control and immunized mice that received one, two, three, or four weekly doses of amoebal extract alone or extract plus CT, and then were challenged with a lethal dose of virulent live amoebae.

All control mice died between days 6–8, while immunized mice with extract alone died between days 7–13. On the other hand, mice that died from the groups immunized with extracts plus CT delayed the mortality for up to days 10 and 13 after the challenge.

Regarding the protection found, we observed that immunization with *N. fowleri* extract plus CT on two occasions, and immunization with extract alone on three occasions, provided 20% protection but were not statistically different (*p* > 0.05) from immunized groups on one occasion or those immunized twice with extract alone. 

In immunized mice on four occasions with extract alone or extract plus CT, we obtained 60% and 100% of protection, respectively, the same protection percentages previously reported by Rojas et al. (2004) [5]. Surprisingly, we observed that immunized mice three times with extract plus CT significantly increased survival up to 80% (*p* < 0.05) compared with those that were immunized on two occasions with both treatments but this was not statistically different from the group immunized with extract plus CT on four occasions (*p* > 0.05). As we can observe, CT delayed the mortality of nonprotected immunized mice and increased the protection percentage compared with immunized mice with extract alone.

### 2.2. Immunogenic Polypeptide Bands of N. fowleri Recognized by IgA from Serum and Nasal Washes

It is worth mentioning that in all Western blot results, both the increase in the immunogenicity of the polypeptide bands and the adjuvant effect of the cholera toxin on the polypeptides was determined by the appearance of new bands or by an increase in the intensity of pre-existing bands.

The reaction of IgA in serum from the control nonimmunized mice group (only received PBS) was not detected (Figure 1A), neither were those that were immunized with extract alone and with extract of *N. fowleri* plus CT for one to three times (data not shown). However, in the fourth immunization, we can observe three faint bands corresponding to 100, 70, and 50 kDa (Figure 1A, blue arrows), where the bands of 70 and 50 kDa were slightly influenced by the adjuvant effect of cholera toxin (Figure 1A, red arrows). However, the adjuvant effect is more evident on the 100 kDa polypeptide band (Figure 1A, red arrow). Two new polypeptide bands corresponding to 37 and 31 kDa were lightly recognized by IgA due to cholera toxin influence (Figure 1A, red arrows). The intensity differences in the polypeptide bands were analyzed by densitometry, where we can clearly observe that there was a significant increase (*p* < 0.001) in the CT adjuvant effect of the analyzed bands (Figure 1B).

Western blot analysis was also performed using nasal washes from control and immunized mice in order to evaluate the local response of the same antibodies.

The results were very similar with respect to those shown in the immunogenic polypeptides of *N. fowleri* recognized by serum IgA, where the reaction of IgA in nasal washes from control nonimmunized mice group was not detected (Figure 1C). Neither were those that were immunized with extract alone nor those immunized with extract of *N. fowleri* plus CT for one to three times (data not shown). However, in the fourth immunization (Figure 1C), we can observe an immunogenic band of 100 kDa (blue arrow), which was also influenced by the adjuvant effect of CT with some other bands such as 70, 58, 55, 50, 48, 40, and 37 kDa (red arrows). The intensity of recognition towards the 100 kDa band was significantly higher (*p* < 0.001) compared to the treatment group of which *N. fowleri* extracts were administered alone (Figure 1D).

### 2.3. Immunogenic Polypeptide Bands of N. fowleri Recognized by IgG from Serum and Nasal Washes

While serum IgG from control mice was totally blank, IgG from all immunized mice recognized a wide spectrum of polypeptides of *N. fowleri* between relative molecular weight (rMW) 19 to 250 kDa. All profiles of polypeptides recognized by serum IgG are shown in Figure 2A. The immunogenic effect, as well as the adjuvant effect of several polypeptides, is observed from the first to the fourth immunization. For example, polypeptide bands that were mainly recognized since the first immunization were those of 70, 60, 58, 50, 40, 37, 31, 29, and 25 kDa (Figure 2A, black arrows), although other polypeptides of higher molecular weight are lightly observed where the intensity of recognition is tenuous (250, 200, 143, 108, and 100 kDa). The adjuvant effect on some polypeptides such as 70, 60, 37, and 25 kDa is observed with stronger labeling, and the presence of new bands, such as 66 and 27 kDa, are also shown (red arrows). In the second immunization, practically, polypeptides that had already appeared since the first immunization are also observed. However, some bands are recognized with much greater intensity, such as 70, 60, and 50 kDa (blue arrows), and approximately nine polypeptide bands were recognized by the IgG antibody due to the adjuvant effect given by CT (red arrows). Immunogenic polypeptides were found, especially in the third and fourth immunization. In the third immunization, the immunogenic effect is clearly observed in polypeptide bands whose molecular weight corresponds to 250, 200, 143, 108, and 100 kDa (blue arrows) and the effect of CT is observed in the bands corresponding to the molecular weights of 100, 92, 50, 37, and 27 kDa (red arrows). Interestingly in the fourth immunization, the immunogenic effect is maintained in bands found in third immunization, but, in addition, this effect increases in polypeptide bands of 100, 58, and 50 kDa. Particularly, there was an increase in recognition intensity in the 100 kDa polypeptide band (blue arrows). The adjuvant effect on 250, 200, 100, 92, 70, 66, 60, 58, 50, 37, 34, and 19 kDa polypeptide bands is clearly notorious. All these polypeptides were found to be more strongly labeled (red arrows). The intensity of some bands was also analyzed (Figure 2B), and significant differences were found in several bands, but we can highlight that the bands of 250, 100, 70, 50, 37, and 19 kDa had especially greater intensity in immunized groups with *N. fowleri* lysates plus CT (particularly, the fourth immunization). When we compared these same bands with the previous immunizations of each treatment, there were also significant differences. This highly significant increase was mainly observed in the polypeptide bands recognized as immunogenic and adjuvanted in third and fourth immunization; these results are summarized in Figure 2B.

Unlike the serum IgG antibody, the quantity of bands recognized by nasal IgG was less abundant. However, we clearly observed an increase in polypeptide recognition as the immunization number is augmented (Figure 2C). After the first and second immunizations, there was no recognition of bands. After the third immunization, a weak immunoblot reaction was obtained with two bands, those of 100 and 70 kDa (Figure 2C, blue arrows). Clearly, we can observe that the immunogenic band of 100 kDa was also influenced by the adjuvant effect of CT (Figure 2C, red arrow). Additionally, the polypeptide bands corresponding to 70, 60, 58, and 50 kDa were lightly detected when CT is used as an adjuvant (red arrows). Finally, by the fourth immunization, in addition to the pre-existing immunogenic polypeptide bands of 100, 70, 60, and 50 kDa, an immunogenic band with an approximate rMW of 31 kDa (blue arrow) also appeared. Again, all these polypeptides were adjuvanted by the CT effect, as well as the two other new bands that were also detected, those of 58 and 37 kDa (red arrows). Polypeptide bands such as 100, 70, 50, and 37 kDa were analyzed densitometrically (Figure 2D). Particularly, 100 and 70 kDa bands were significantly higher in terms of intensity compared to the rest of the groups.

### 2.4. Immunogenic Polypeptide bands of N. fowleri Recognized by IgM from Serum 

A broad pattern of *N. fowleri* polypeptides recognized by serum IgM was also found (Figure 3A), most of which ranged from 19 to 250 kDa. Apparently, no dramatic change regarding immunogenicity is found, but analyzing the first immunization, we can observe an adjuvant effect on polypeptides that ranged from 25 to 58 kDa (Figure 3A, red arrows). For the second immunization, the immunogenicity is increased for polypeptides with a molecular weight of 143, 108, 60, 58, 52, 50, 48, 40, 31, 29, 27, and 25 kDa (blue arrows), and the adjuvant effect was especially on polypeptides corresponding to 250, 200, and 58 kDa (red arrows). After the third and fourth immunization, the polypeptide profile is very similar to that found of the second immunization. However, proteins such as 250, 200, 108, 100, 58, 52, 50, 48, 40, 37, 34, 31, 29, 27, 25, and 19 kDa had been remarkably influenced by the adjuvant effect of CT (red arrows), while in the fourth immunization, the immunogenic effect is found in polypeptides from 19 to 250 kDa, where a 70-kDa band appears (blue arrows). An adjuvant effect influenced by CT was also found on bands of 108, 100, 70, 34, 27, and 19 kDa (red arrows). Finally, densitometric analysis details of 250, 100, 70, 50, 37, and 19 kDa bands are shown in Figure 3B.

It is worth mentioning that mucosal IgM (mouse nasal washes) did not recognize any antigen from *N. fowleri* (data not shown).

Among the immunogenic bands recognized by the antiserum and nasal washes IgA, IgG, and IgM, we highlight those with a relative molecular weight of 250, 100, 70, 50, 37, and 19 kDa. Most of these immunogenic polypeptide bands were adjuvanted by CT and consistently recognized by all analyzed isotypes, predominating a major intensity in the third and fourth immunization with *N. fowleri* extract plus CT.

### 2.5. Identification of Immunogenic Polypeptide Bands by Mass Spectrometry

Based on immunoblot results, the selected immunogenic polypeptide bands were identified by nanoliquid chromatography tandem mass spectrometry (nanoLC–MS/MS) on the basis of peptide mass matching, following in-gel digestion with trypsin. As the genome of *N. gruberi* was sequenced in 2010 [16], and because *N. fowleri* and *N. gruberi* share substantially more gene families than *N. fowleri* shares with the other species [17], the peptide masses were matched with the theoretical peptide masses of all proteins from both *Naegleria* species (*N. fowleri* and *N. gruberi*) on the UNIPROT database. Therefore, polypeptides of the immunogenic bands were successfully identified as peptide masses were matched with the theoretical peptide of *N. fowleri*, and peptides that did not match with *N. fowleri* were found to match with *N. gruberi* (Table 2).

Some proteins that matched with *N. fowleri* and *N. gruberi* polypeptides are described below and shown in Table 2, along with other proteins. The details of the mass spectrometry results can be found in Appendix A.

Particularly, for *N. fowleri*, myosin II heavy chain (Q25561) was found in three polypeptide bands, those with a relative molecular weight of 250, 100, 50, and 37 kDa. The peptides present in 100, 70, 50, and 19 kDa bands were matched with a membrane protein (Q95UJ2), whereas peptides from bands corresponding to molecular weights of 250, 100, and 37 kDa were matched with actin (B5M6J9). Peptides corresponding to the heat shock protein 70 family (Q6B3P1) were identified in three bands, 100, 70, and 50 kDa. Two polypeptide bands, 100 and 50 kDa, contained peptides that matched for amino acid decarboxylase (C6L6E3). Pyrophosphate fructose 6 phosphate 1 phosphotransferase (PFP) was matched with peptides present in the 50 kDa band. On the other hand, a putative uncharacterized protein (D2UXB7) of *N. gruberi* was identified in three bands (those corresponding to 100, 50, and 37 kDa), whereas a glyceraldehyde 3 phosphate dehydrogenase (D2W142) was found in 100 and 37 kDa polypeptide bands. Finally, another enzyme, such as protein disulfide isomerase (D2VCZ2), was matched with the 50 kDa polypeptide band. 

## 3. Discussion

The humoral response plays an important role against *Naegleria fowleri* meningoencephalitis in mice [5,6,7]. Therefore, the identification of antibody-accessible proteins on the pathogen cell is important in the selection of vaccine candidates. In previous studies, we immunized mice with *N. fowleri* lysates plus cholera toxin (CT) or Cry1Ac, and we showed the production and secretion of IgA and IgG antibodies as well as a considerable number of polymorphonuclear cells (PMNs), suggesting these mechanisms avoid the attachment of *N. fowleri* to the apical side of the nasal epithelium and subsequent invasion of the amoeba [5,6,18]. However, the actual immunogens that elicit the protective immune response against *N. fowleri* meningoencephalitis in mice are unknown. Only a 13.1-kDa protein named Nfa1 has been previously reported, which, when adjuvanted with CTB, is able to induce strong protective immunity in mice with PAM [9].

The present study focuses on the identification of some immunogenic proteins from *N. fowleri* through Western blot analyses probed with antiserum and nasal washes from immunized mice, with total extracts of *N. fowleri* plus cholera toxin (CT) after one, two, three, or four immunizations. This is the first time that polypeptide band profiles of *N. fowleri* using these antibodies are reported. Although some studies have reported the virulent protein profiles from *N. fowleri* [19], they have not considered the immunogenic protein bands or spots recognized by the antiserum and nasal washes of IgA, IgG and IgM from immunized and protected mice.

As we mentioned above, in a recently published work [11], we found differential proteins between *N. fowleri* and *N. lovaniensis*, specifically using IgG antibodies by 2-DE Western blot assay. However, the analysis did not include an immunization scheme, with or without CT as an adjuvant, in order to assess survival rates with different numbers of immunizations. Moreover, the specific protein profile given by the different immunoglobulins was not evaluated.

In the present study, we observed that the major antibody reactivity was directed to several polypeptide bands, such as 250, 100, 75, 50, 37, and 19 kDa, especially in the third and fourth immunizations. Furthermore, these polypeptide bands were detected consistently by all immunoglobulins (IgA, IgG, and IgM) in both serum and nasal washes. Interestingly, many immunogenic bands, including these six bands, were adjuvanted by CT, where some proteins were identified, such as myosin II heavy chain (Q25561), actin (B5M6J9), ATP synthase F1 subunit alpha (M4H5H9), amino acid decarboxylase (C6L6E3), membrane protein (Q95UJ2), Hsp70 (Q6B3P1), and glyceraldehyde 3 phosphate dehydrogenase (D2W142). These are proteins that had been previously reported by Gutiérrez-Sanchez et al. (2020) [11], with differential expression between *N. fowleri* and *N. lovaniensis*. 

In previous studies, actin and myosin have particularly been reported to participate in the phagocytosis process, suggesting that both proteins are concentrated around phagocytic cups of *N. fowleri* [20]. *N. fowleri*-actin is found in the pseudopods, cytoplasm, and especially in structures known as "food-cups" (amoebastome), which can play a significant role in the invasion process of the amoeba, particularly in the phagocytosis process [21]. Recently, an *nf*-actin gene was cloned, which expressed a 50.1-kDa recombinant protein (*Nf*-actin). In that work, the authors found that the overexpression of this recombinant protein in *N. fowleri* had an increase in adhesion activity towards extracellular matrix components, fibronectin, collagen I, and fibrinogen, consequently showing an increase in phagocytosis and cytotoxicity compared with wild-type *N. fowleri* [22]. Other studies have shown that actin transcript levels were slightly elevated in the highly virulent *N. fowleri* trophozoites compared to weakly virulent amoebae [23]. On the other hand, regarding some enzymes identified in these immunogenic bands, such as ATP synthase F1 subunit alpha, amino acid decarboxylase and glyceraldehyde 3 phosphate dehydrogenase, they are also known as housekeeping enzymes that are ubiquitous in almost all living beings in order to carry out fundamental metabolic functions for survival. However, in recent years, there have been several studies indicating that some pathogens use these enzymes to increase their virulence. Consequently, this type of enzymes could act as targets for the design of novel strategies to control infections by using agents that can inhibit or block their action [24], and, thus, may also serve as candidates for vaccine development against PAM. 

In the protection model, these immunogenic polypeptides may probably stimulate a neutralization response to *N. fowleri* trophozoites, which might be good targets for vaccine development. In a future study, it would be necessary to investigate the immune protective effects through the purification of these immunogenic polypeptides bands to immunize and challenge mice. We could consider performing neutralization assays using specific antibodies for each candidate protein.

It is known that proteins with immunogenicity may stimulate the host to produce various degrees of neutralization, and only those proteins with high neutralization ability can become targets for vaccine candidates [25,26].

It is worth mentioning that the peptides that did not match *N. fowleri* have not yet been reported for this species. Thus, comparing the peptides sequences with a nonpathogenic species (*N. gruberi*) would lead us, in future studies, to find information from proteins reported for other pathogens as virulence factors. Such proteins may be the basis for the design of synthetic peptides based on these proteins. An example could be the glyceraldehyde-3-phosphate dehydrogenase that was identified in the immunogenic bands of 100 and 37 kDa. This protein has been reported in several studies as a virulence factor of pathogenic microorganisms, and it has also been used as a therapeutic target for the design of vaccines against the diseases caused by these pathogens [15,24,27,28,29]. 

The antigen recognition of the antibodies, mainly IgG and IgM from serum, was directed toward many polypeptide bands from 19 to 250 kDa. The recognition intensity toward several bands was higher by IgG, where the 250, 100, 70, 50, 37, and 19 kDa bands were clearly observed in the fourth immunization. In the present study, we only selected the previously mentioned immunogenic bands for the mass spectrometric analysis due their consistency in nasal washes, although other important bands were also observed, which will have to be analyzed in further work.

Although it is known that secretory IgA is the principal antibody against pathogens and toxins that might penetrate mucosal surfaces [30], in our study, the recognition of several antigens, mainly by IgG of serum in the third and fourth immunization, and by IgG of nasal washes in the fourth immunization, led us to suggest that this immunoglobulin might also be correlated with protection against *N. fowleri* infection. This is because nasal mucosa IgG could be derived from the plasma by process of passive transudation along a concentration gradient, as reported by Wagner et al. (1987) [31]. These authors demonstrated that IgG was not locally produced; however, this immunoglobulin was the major contributor to resistance in the nasal compartment of humans vaccinated against influenza A. They also suggested that the magnitude of this gradient might have important implications for the protection of mucosal surfaces by systemic antibodies. 

In addition to this, in our working group, it was reported that after nasal immunization with *N. fowleri* lysates and Cry1Ac or CT, olfactory epithelium became more permeable, and more IgG leaks into the secretions in this way, limiting nasal *N. fowleri* adhesion together with IgA [6,18]. Thus, the polypeptides recognized in the present work by these antibodies could represent new candidates for the design of a vaccine against PAM.

Regarding the adjuvanticity of CT, we observed the effect on some polypeptide bands, where the recognition of the antibodies is increased when the mice are immunized with *N. fowleri* lysate plus cholera toxin. The induction of antibody responses by CT is well established and antigens coadministered with CT or CTB as adjuvant strongly potentiate the immunogenicity of most of the antigens that are given at the same time and at the same mucosal surface. Particularly, CT has been shown to (a) induce an increase in the permeability of the mucosal epithelium, leading to improved uptake of coadministered antigenic proteins, (b) improve antigen presentation by different antigen-presenting cells (APCs), and (c) contribute isotype differentiation in B-cells, resulting in increased antibody formation [32,33,34]. Related to this, we had already demonstrated that the adjuvant effect, given by CT on *N. fowleri* lysates, influences the increase of macrophages and dendritic cells, and therefore, protective immunity would depend on interactions between APCs and T-cells from nasal passages and NALT [8].

It would be interesting if some of the identified peptides or proteins were coadministered with CT or with non-toxic CTB subunit, as reported for Nfa1, where it was found that the effect of protective immunity of the Nfa1 protein for *N. fowleri* infection was enhanced by using CTB [9]. 

In conclusion, further studies are needed to determine the role of these proteins and their biological functions in PAM disease, using transcriptomic and proteomic analyses in order to contribute to the generation of vaccines that include epitopes or antigenic determinants sufficient to activate the appropriate cellular and humoral protective responses against *N. fowleri*.

## 4. Materials and Methods 

### 4.1. Naegleria fowleri Cultures

*N. fowleri* ATCC 30,808 trophozoites were cultured in axenic conditions at 37 °C in bactocasitone broth (Difco, Le Pont de Claix, France), supplemented with 10% bovine fetal serum (Gibco, Grand Island, NY, USA). Trophozoites were chilled and harvested during the logarithmic phase (72 h). After centrifuging at 1500× *g* for 10 min, amoebas were washed 3 times with phosphate-buffered saline (PBS) and counted in a hemocytometer. The virulence of *N. fowleri* was reactivated by serial passage through mice, as described previously [5]. Only freshly recovered virulent amoebae were used for all experiments.

### 4.2. Animals

All procedures performed in this study, which involved BALB/c mice, were in accordance with the Mexican federal regulations for animal experimentation and care (NOM-062-ZOO-1999, Ministry of Agriculture, Mexico City, Mexico) and approved by the ethical standards of the Institutional Animal Care and Use Committee (Number of Approval ESM-CICUAL-ADEM-05/27-09-2019). In all experiments, 6- to 8-week-old male BALB/c mice were used. These mice were maintained in horizontal laminar flow cabinets and provided food and water ad libitum.

### 4.3. Immunization Scheme

Five experimental groups of 30 mice each were intranasally immunized as follows: (1) one immunization, (2) two immunizations, (3) three immunizations, and 4) four immunizations. Briefly, 15 animals of each group were immunized with amoebic lysates (100 μg of protein) plus 2 μg of cholera toxin (CT; Sigma-Aldrich), whereas the other 15 animals were immunized with amoebic lysates alone (100 μg of protein). After each immunization, mice were challenged with 5 × 10^4^ live virulent *N. fowleri* trophozoites in 30 μL of PBS. After 24 h of the challenge, 5 mice from each treatment group were sacrificed and pooled serum and nasal washes were obtained. The 10 mice remaining in each treatment group were used to assess survival after the challenge. Control mice received 30 µL of PBS.

### 4.4. Sample Collection

Serum samples were obtained from blood extracted by cardiac puncture from isoflurane-anesthetized mice. Nasal fluids were also collected from some mice. Exsanguinated mice were decapitated, the heads were rinsed with ice-cold phosphate-buffered saline (PBS) to remove blood, a polyethylene tube was inserted via the oropharynx into the nasopharyngeal cavity, and contents of the nasal passages from individual mice were washed out of the nares with 1 ml of PBS. Samples that became contaminated with blood during collection were discarded.

### 4.5. Anti-N. fowleri Antibodies by Western Blot

Trophozoites of *N. fowleri* were lysed by sonication with one 10-s pulse at 100 W of amplitude (Fisher Sonic Dismembrator model 300). The resulting suspension was centrifuged at 8 °C for 1 h at 4000× *g*, and then the supernatant was eliminated, and the pellet resuspended in 3 mL of PBS containing 5 mM p-hydroxymercuribenzoic acid (Sigma Chemical Co., St. Louis, Mo.). The protein concentration was quantified by the Bradford method.

For the detection of immunogenic polypeptides, the *N. fowleri* extracts were electrophoresed on 10% sodium dodecyl sulfate polyacrylamide gel electrophoresis (SDS–PAGE) and were then transferred to a nitrocellulose membrane [35]. The membranes were blocked by incubation with PBS buffer pH 7.4, containing 10% nonfat dry milk for 1 h; subsequently, the strips were washed with 0.05% Tween-20 in PBS (PBS-T) and incubated overnight at 4 °C with serum (1:100) and nasal washes from the different groups of mice. The membrane was washed with PBS-T for 10 min with agitation and incubated overnight at 4 °C, with radish peroxidase-labeled goat anti-mouse IgG, IgA, and IgM antibodies (Zymed Laboratories, San Francisco, Calif.), diluted 1:1000. Finally, the recognized proteins were revealed with a substrate solution (H2O2,3.6 mM 4-chloro-1-napthol; Pierce).

### 4.6. Identification of Immunogenic Bands by Mass Spectrometry

Immunogenic proteins bands recognized by the antibodies were matched with the corresponding protein bands on images of stained Coomassie Blue gel to excise the protein bands of interest. After comparison with immunoblotting results, six polypeptide bands were manually excised from gels and then sent for protein identification analysis by nanoLC–ESI-MSMS (USAI-Facultad de Química-UNAM).

The MSMS peptide mass data were automatically searched against the UniProt protein database (https://www.uniprot.org/blast/) using the global ProteinLynx version 2.4 server and software (Waters Corporation), with a Protein Lynx Global Server (PLGS) (Waters Corporation). A PLGS score of >95% confidence was accepted as correct. The peptides were matched with the theoretical peptides of reported proteins from *N. fowleri* and *N. gruberi*.

### 4.7. Survival Analysis, Densitometric Analysis by ImageLab and Statistical Analysis

The survival analysis was performed using a comparison of survival curves and log-rank (Mantel–Cox) test for survival assays with the PRISM computer program (GraphPad, San Diego, Calif.). *p* < 0.05, *p* < 0.01, or *p* < 0.001 was considered statistically significant. Protein bands of interest were scanned with the GS-900 calibrated densitometer (Biorad). The ImageLab program was used to establish changes in the densitometry of the Western blot immunogenic bands corresponding to the treatment and the different number of immunizations. Data represent mean percentages ± SD of three independent experiments (bands of three independent repetitions of Western blot). The data obtained were statistically analyzed using *t*-tests (and nonparametric tests), and then an unpaired *t*-test. A level of significance with *p* < 0.05, *p* < 0.01, or *p* < 0.001 was considered to establish significant differences between compared immunogenic bands.

## Figures and Tables

**Figure 1 pathogens-09-00460-f001:**
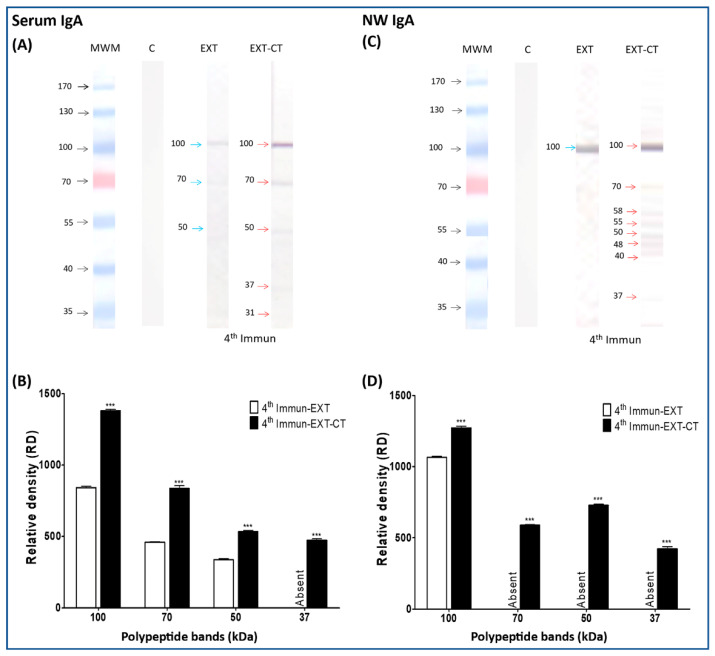
Detection of immunogenic polypeptide bands of *N. fowleri* recognized by IgA from serum and nasal washes. *N. fowleri* total extracts were electrophoresed on 10% SDS–PAGE and were transferred to a nitrocellulose membrane and incubated with serum and nasal washes from the different groups of immunized mice (different number of immunizations; 1, 2, 3, or 4), with *N. fowleri* extract alone or with amoebic extract of *N. fowleri* plus CT, and then incubated with radish peroxidase-labeled goat anti-mouse (**A**,**C**). Polypeptide bands (100, 70, 50, 37, and 19 kDa) were analyzed densitometrically using ImageLab software (**B**,**D**). Data represent mean percentages ± SD of three independent experiments (bands of three independent repetitions of each immunoblot). A level of significance with **p* < 0.05, ***p* < 0.01 or ****p* < 0.001 was considered to establish a significant difference in the same relative molecular weight (rMW) of the bands between two treatment groups (extract alone or extract plus CT, with the same number of immunizations). Immunizations (Immun), extract (EXT), cholera toxin (CT), nasal washes (NW), molecular weight marker (MWM). Absent: the polypeptide band was not detected.

**Figure 2 pathogens-09-00460-f002:**
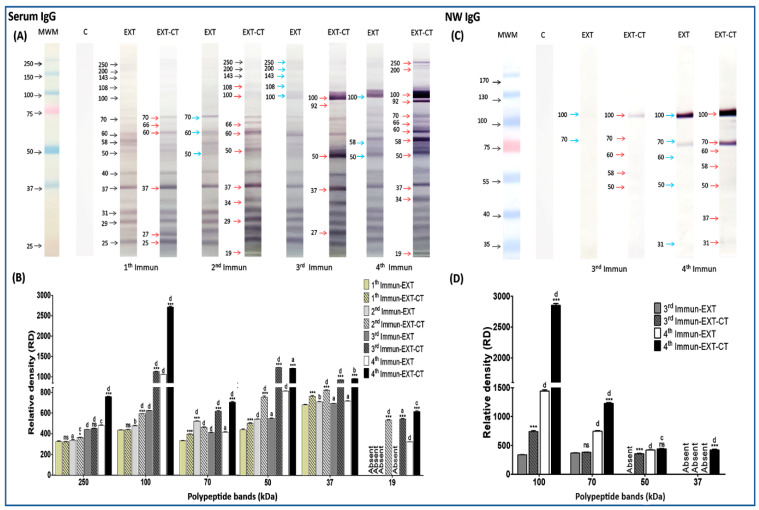
Detection of immunogenic polypeptide bands of *N. fowleri* recognized by IgG from serum and nasal washes. *N. fowleri* total extracts were electrophoresed on 10% SDS–PAGE and were transferred to a nitrocellulose membrane and incubated with serum and nasal washes from the different groups of immunized mice (different number of immunizations; 1, 2, 3, or 4), with *N. fowleri* extract alone or with amoebic extract of *N. fowleri* plus CT, and then incubated with radish peroxidase-labeled goat anti-mouse (**A**,**C**). Polypeptide bands (250, 100, 70, 50, 37, and 19 kDa) were analyzed densitometrically using ImageLab software (**B**,**D**). Data represent mean percentages ± SD of three independent experiments (bands of three independent repetitions of each immunoblot). A level of significance with **p* < 0.05, ***p* < 0.01 or ****p* < 0.001 was considered to establish a significant difference in the same relative molecular weight (rMW) of the bands between two treatment groups (extract alone or extract plus CT, with the same number of immunizations). No statistically significant difference (ns). A level of significance with bP < 0.05, cP < 0.01, or dP < 0.001 was considered to establish significant difference, comparing each treatment group with the previous immunization. No statistically significant difference (a). Immunization (Immun), extract (EXT), cholera toxin (CT), nasal washes (NW), molecular weight marker (MWM). Absent: the polypeptide band was not detected.

**Figure 3 pathogens-09-00460-f003:**
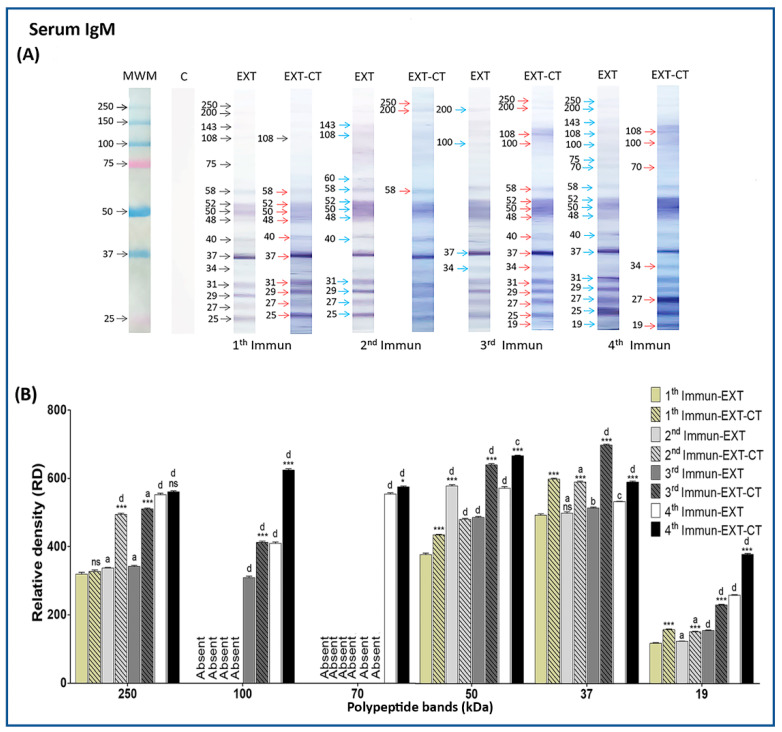
Detection of immunogenic polypeptide bands of *N. fowleri* recognized by IgA from serum and nasal washes. *N. fowleri* total extracts were electrophoresed on 10% SDS–PAGE and were transferred to a nitrocellulose membrane and incubated with serum and nasal washes from the different groups of immunized mice (different number of immunizations; 1, 2, 3, and 4), with *N. fowleri* extract alone or with amoebic extract of *N. fowleri* plus CT, and then incubated with radish peroxidase-labeled goat anti-mouse (**A**). Polypeptide bands (250, 100, 70, 50, 37, and 19 kDa) were analyzed densitometrically using ImageLab software (**B**). Data represent mean percentages ± SD of three independent experiments (bands of three independent repetitions of each immunoblot). A level of significance with **p* < 0.05, ***p* < 0.01 or ****p* < 0.001 was considered to establish a significant difference in the same relative molecular weight (rMW) of the bands between two treatment groups (extract alone or extract plus CT, with the same number of immunizations). No statistically significant difference (ns). A level of significance with bP < 0.05, cP < 0.01, or dP < 0.001 was considered to establish significant difference comparing each treatment group with the previous immunization. No statistically significant difference (a). Immunization (Immun), extract (EXT), cholera toxin (CT), nasal washes (NW), molecular weight marker (MWM). Absent: the polypeptide band was not detected.

**Table 1 pathogens-09-00460-t001:** Survival and protection.

Treatment	Mice Death (in days)	% Protection
Control	6–8	0
1st immun-ext	7–11	0
1st immun-ext-CT	10–13	0
2nd immun-ext	8–12	0
2nd immun-ext-CT	11–13	20
3rd immun-ext	11–13	20
3rd immun-ext-CT	13	80
4th immun-ext	11–13	60
4th immun-ext-CT	--	100

Different groups of BALB/c mice (10 per group) were immunized and challenged with 5 × 10^4^ live virulent *N. fowleri* trophozoites in 30 μL of PBS. Survival rate was determined after the challenge. Animals were monitored for up to 60 days. Control mice received 30 µL of PBS. Immun: immunization. Ext: extract. CT: cholera toxin.

**Table 2 pathogens-09-00460-t002:** Identification of immunogenic polypeptide bands of *N. fowleri* by mass spectrometry.

Immunogenic Polypeptide Bands (kDa)	Protein Description with the Compared Species	
	*Naegleria fowleri*	*Naegleria gruberi*
**250**	Myosin II heavy chain Fragment **(Q25561)**; Actin Fragment **(B5M6J9)**; Actin 1 **(ACT1)**, Unknown protein NF009 from 2D PAGE Fragment **(NF09)**; Cpn 60 Fragment **(Q94626)**; Ubiquitin Fragment **(Q25558)**.	Cysteine protease Fragment **(Q9TWP8)**.
**100**	Actin Fragment **(B5M6J9)**; Ubiquitin Fragment **(Q25558)**; Heat shock protein **(Q25552)**; Membrane protein **(Q95UJ2)**; Amino acid decarboxylase Fragment **(C6L6E3)**; 26S proteasome subunit Fragment **(Q95VC2)**; Penicillin amidase homolog Fragment **(Q25548)**; ATP synthase subunit alpha **(M4H5H9)**; Myosin II heavy chain Fragment **(Q25561)**; Hsp70 **(Q6B3P1)**; Ribosomal protein S3 **(M4H5R4)**; Putative cytosolic carboxypeptidase 6 **(M1H4M8)**.	Putative uncharacterized protein **(D2UXB7)**; Predicted protein **(D2VY23)**; Predicted protein **(D2W5T0)**; Predicted protein **(D2VU93)**; Polyubiquitin **(D2VTG7)**; Calponin homology domain protein Fragment **(D2V120)**; Predicted protein Fragment **(D2UXT6)**; Glyceraldehyde 3 phosphate dehydrogenase **(D2W142)**; Glutamate dehydrogenase **(D2V2V3)**; AP complex subunit beta **(D2VMT9)**; Catalase **(D2VWQ6)**; Dihydroorotate dehydrogenase family protein **(D2V5Y1)**; ATP synthase subunit beta **(D2V5T0)**.
**70**	Actin 1 **(ACT1)**; Actin 2 **(ACT2)**; Membrane protein **(Q95UJ2)**; Hsp70 **(Q6B3P1)**.	Predicted protein **(D2UZ33)**; Predicted protein **(D2VWJ8)**; Predicted protein **(D2VSC1)**; Methylcrotonyl CoA carboxylase **(D2VDR3)**; Predicted protein **(D2V9C6)**; AP complex subunit beta **(D2VMT9)**; Clathrin heavy chain **(D2V5V4)**; ATP synthase subunit beta **(D2V5T0)**; Predicted protein **(D2VGK9)**; Isovaleryl CoA dehydrogenase **(D2UXG9)**; Predicted protein **(D2VWE1)**.
**50**	Amino acid decarboxylase Fragment **(C6L6E3)**; Actin 1 **(ACT1)**; Pyrophosphate fructose 6 phosphate 1 phosphotransferase **(PFP)**; Membrane protein **(Q95UJ2)**; Thioredoxin homolog **(Q25549)**; Hsp70 **(Q6B3P1)**; Myosin II heavy chain Fragment **(Q25561)**.	Glutamate dehydrogenase **(D2V2V3)**; Putative uncharacterized protein **(D2UXB7)**; Predicted protein **(D2VT95)**; Predicted protein **(D2VWF2)**; Pyrophosphate fructose 6 phosphate **(D2V5S29)**; Predicted protein **(D2V0Z9)**; Adenosylhomocysteinase **(D2W1E4)**; Coronin **(D2VLM1)**; Elongation factor Tu **(D2V3J1)**; Predicted protein **(D2VV89)**; ATP synthase subunit beta **(D2V5T0)**; Predicted protein **(D2V9Y7)**; Alanine aminotransferase **(D2V3R0)**; Predicted protein **(D2W0R2)**; Predicted protein **(D2W1W8)**; Elongation factor 1 alpha Fragment **(Q2MM01)**; Protein disulfide isomerase **(D2VCZ2)**; Serine hydroxymethyltransferase **(D2UYA5)**; Rab GDP dissociation inhibitor **(D2V193)**; Actin related protein ARP3 **(D2V7J7)**; RhoGEF domain containing protein **(D2VJF9)**; Predicted protein **(D2UXY9)**; Succinate CoA ligase ADP forming subunit beta mitochondrial **(D2VP45)**; 4 aminobutyrate aminotransferase **(D2UY50)**; Predicted protein **(D2W3B8)**; Dihydrolipoamide succinyltransferase **(D2V1E8)**; Predicted protein **(D2VB42)**; Predicted protein **(D2VBF6)**; Predicted protein **(D2VLT8)**; Predicted protein **(D2UZW5)**; Predicted protein **(D2W2M7)**.
**37**	Actin 2 Fragment **(ACT2)**; Actin Fragment **(B5M6J9)**; ATP synthase subunit alpha **(M4H5H9)**; Ubiquitin Fragment **(Q25559)**; Myosin II heavy chain Fragment **(Q25561)**.	Putative uncharacterized protein **(D2UXB7)**; Putative uncharacterized protein **(D2VE39)**; Putative uncharacterized protein **(D2VZJ8)**; Conventional actin **(D2VUS6)**; Glyceraldehyde 3 phosphate dehydrogenase **(D2W142)**; Putative uncharacterized protein **(D2V8I2)**; Predicted protein (D2VM36); Predicted protein **(D2VKN7)**; Mitogen activated protein kinase **(D2VK26)**; Predicted protein **(D2VSP0)**; Malate dehydrogenase **(D2VZB1)**; Prohibitin **(D2W2C5)**; Isocitrate dehydrogenase NAD subunit mitochondrial **(D2VQ70)**; Isocitrate dehydrogenase NADP dependent **(D2VRD7)**; Mitochondrial trans-2 enoyl CoA reductase **(D2VNA9)**; Inosine 5 monophosphate dehydrogenase **(D2UY23)**; ATP synthase subunit beta **(D2V5T0)**; Predicted protein **(D2VKI4)**; ARF SAR family small GTPase **(D2V115)**; Jun kinase activation domain binding protein **(D2VD78)**; Predicted protein **(D2VRL5)**; Elongation factor 1 alpha Fragment **(Q2MM01)**; NAD dependent epimerase dehydratase family protein **(D2V803)**; Putative uncharacterized protein **(D2VEC2)**.
**19**	Membrane protein **(Q95UJ2)**; Unknown protein NF016 from 2D PAGE Fragment **(NF16)**; Calcineurin B **(Q9NAY9)**; Fructose 1 6 bisphosphatase homolog Fragment **(Q27706)**; Photoactivated adenylyl cyclase **(W0SLB1)**.	40S ribosomal protein S13 **(D2VL67)**.

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
