# Peer review of "Identification of Immunogenic Antigens of *Naegleria fowleri* Adjuvanted by Cholera Toxin"

_pathogens, 2020, doi:10.3390/pathogens9060460_

Round 1
Reviewer 1 Report
This is a report for the manuscript entitled “Identification of immunogenic antigens of Naegleria fowleri adjuvanted by cholera toxin”.
The manuscript presents and evaluates a possible progress towards a vaccine against an acute and fatal disease of the central nervous system called primary amoebic meningoencephalitis (PAM). The subject was addressed by adequate methodological approaches, which rendered relevant results.
However, the manuscript still deserves a global revision to make it more clear: the figures do not have enough resolution and many sentences need be revised/written, as indicated in Specific Comments.
Specific comments
Abstract
Sentence in lanes 23-25: “a western blot…” of what? What polypeptides?
Line 30: “Some interestingly proteins identified in this study…” This an example of an expression needing revision.
Introduction
The Introduction is full of sentences needing revision as, for example: lines 49-53, 53-56, 8-88, and 88-91
Line 56: “dendritic cell” OR “dendritic cells”?
Moreover, the last paragraph (lines 92-99) of the Introduction is supposed to formulate the objectives of the study, NOT their Conclusions!
Results
First of all, a general observation: please consider, along the Results section, the substitution of the word “protein” for “polypeptide”. These results refer to SDS-PAGE electrophoresis that resolves polypeptides due to its denaturant conditions.
Paragraph in lines 102-117 is composed of sentences needing revision, most of them not related to the results themselves. Some sentences would be more appropriate for the Introduction or for the Discussion. The results (Figures 2 and 3 and Table 1) should be presented and explained, without a discussion...
Lines 149-151: this first sentence of the legend is not so clear as the following sentence is, which says exactly the same! Please remove the first one. Figure 1 includes a table, this one more informative than the image (graphic). Consider the presentation of only one of them: the clearer and more informative.
Figures 2 and 3: Bands and annotations inside the figures without enough resolution. This aspect did not allow a complete verification of the results described, yet it appears correct and adequate. This aspect should be considered to improve the recognition of the importance of these results. Moreover, a line without bands, corresponding to the controls, would be adequate to show as well!
Lines 284-286: another example of an incomplete and not understandable sentence. Maybe without the period in the end of this sentence…we could understand its message.
Table 1: It could be considered to move it for Supplementary data, due to its extension and scope.
Discussion
As referred for other sections of the manuscript, a careful revision of the discussion would be necessary, to make it more comprehensible and valuable.
Line 336: Missing the reference to the 119 kDa band.
Line 385-386: Another sentence that could make sense without the period, since its conclusion appears in the next sentence.
Lines 408-412: A long sentence difficult to understand.
Line 418: Reference to “new” vaccines… In this case, a reference to previous vaccines is missing either in the Introduction or in the Discussion.
Materials and Methods
Line 436: How animals were immunized, is missing.
Reviewer 2 Report
Manuscript (Pathogens-805913) submitted by Carrasco-Yépez et al. “Identification of immunogenic antigens of Naegleria fowleri adjuvanted by cholera toxin” discussed and presented the work related to a neglected tropical pathogen N. fowleri. This occasional amoeba species causes meningoencephalitis and author/s has tried to detect and identify the immunogenic antigens which could be candidate target for its protection.
Although author/s has done good work but there are several issues/errors which makes this manuscript weak. I am here highlighting few of them, as list are very long and assuming that author will take of rest.
From the beginning, in abstract as authors has written “we previously reported” and it is also look like similar work to their previous report of 2014 paper published in Experimental Pathology “Intranasal Coadministration of Cholera Toxin With Amoeba Lysates Modulates the Secretion of IgA and IgG Antibodies, Production of Cytokines and Expression of pIgR in the Nasal Cavity of Mice in the Model of Naegleria Fowleri Meningoencephalitis”. It is recommended that abstract and work should be more focused on the present work. Use the word present work not recent work (Line no. 53, 68. 330)
Still, finding of above-mentioned paper and present work is same. But, the only new thing here is identification of immunogenic protein bands by mass spectrometry. Then here again, a paper from same group of authors was published in 2020 Parasite Immunol. “Identification of Differential Protein Recognition Pattern Between Naegleria Fowleri and Naegleria Lovaniensis”. What is new in this paper please clarify?
The list of identified protein is very long and presentation of all the data in the main text file is meaningless. I will recommend provide the list only few which you found on protein gel and remaining can be accessible through the supporting materials. It is highly recommended to emphasize the novelty of present work (in abstract and conclusion).
There are N number of minor mistakes, which need to be taken care off. Here are some examples.
- Introduction (Line no. 39) begin with N. Fowleri and Line no. 45 used Naegleria Fowleri.
- Similarly, CTB and LTB (line No. 62) has been abbreviate here and elucidated in line no. 321 first time.
- References- (Wagner et al. 1987) in the text, it is 29 but in bibliography is at 30 not 29.
- Relative density should be abbreviated as RD not DR
- Figure 1, significance lines should be separated from each other. Increase the font size of other figures.
- Spelling and Grammar mistakes are very prevalent. Take care of that.
- Authors contribution should be condensed according to the authors name like MMCY.
I hope author will consider all the suggestions and improvise the manuscript accordingly and from their end too.
All the Very Best.
Reviewer 3 Report
The manuscript written by Hernandez et al is about protein analysis for the identification and identification of immunogenic antigens of N. fowleri
Minor
I suggest to reformulate the abstract and write more clearly the background, aim of the study, experimental part and conclusions
Materials and Methods: authors should report the animal permit for the in vivo procedure. Also which kind of mice have been used? What was the route of administration? A scheme related to the animal experiment would be of help for readers
Round 2
Reviewer 2 Report
Dear Author,
I appreciate that you incorporated all the suggestions in the manuscript and provide the justified answers to the comments.
All the Best.
Thanks